# The Unseen—An Investigative Analysis of Thematic and Spatial Coverage of News on the Ongoing Refugee Crisis in West Africa

Hansi Senaratne [1,*] , Martin Mühlbauer [1] , Ralph Kiefl [1] , Andrea Cárdenas [2] , Lallu Prathapan [2] , Torsten Riedlinger [1] , Carolin Biewer [3] and Hannes Taubenböck [1,2]

1    Earth Observation Center, German Aerospace Center, 82234 Wessling, Germany
2    Institute of Geography and Geology, University of Würzburg, 97074 Würzburg, Germany
3    Department of English and American Studies, University of Würzburg, 97074 Würzburg, Germany
*    Correspondence: hansi.senaratne@dlr.de

**Abstract:** The fastest growing regional crisis is happening in West Africa today, with over 8 million people considered persons of concern. A culmination of identity politics, climate-driven disasters, and extreme poverty has led to this humanitarian crisis in the region and is exacerbated by a lack of political will and misplaced media attention. The current state of the art does not present sufficient investigations of the thematic and spatial coverage of news media of this crisis in this region. This paper studies the spatial coverage of this crisis as reported in the media, and the themes associated with those locations, based on a curated dataset. For the time frame 12 March to 15 September 2021, 2017 news articles related to the refugee crisis in West Africa were examined and manually coded based on (1) the geographical locations mentioned in each article; (2) the themes found in the articles in reference to a location (e.g., *Relocation* of people in *Abuja*). The dataset introduces a thematic dimension, as never achieved before, to the conflict-ridden areas in West Africa. A comparative analysis with UNHCR (United Nations High Commissioner for Refugees) data showed that 96.8% of refugee-related locations in West Africa were not covered by news during the considered time frame. Contrastingly, 80.4% of locations mentioned in the news do not appear in the UNHCR repository. Most news articles published during this time frame reported on *Development aid* or *Political statements*. Linear multiple regression analysis showed *GDP per capita* and *political stability* to be among the most influential determinants of news coverage.

**Keywords:** West African refugee crisis; news media reporting; spatio-thematic coverage

## 1. Introduction

According to a report by the World Health Organization (Available online: https://www.who.int/emergencies/situations/humanitarian-crisis-in-sahel-region-of-africa (accessed on 1 March 2023)), over 33 million people from the Sahel region of West Africa were in need of urgent care in the year 2022 alone as a result of the ongoing conflict, giving rise to one of the *fastest growing regional crises in the world*. Many regionalized conflicts along state borders in North and West Africa are driven by *identity politics*, which has caused militant groups such as the Al-Qaeda or Boko Haram to evolve in these regions in recent years [1]. The crisis in the Sahel region began in 2011 when armed groups took control over Northern Mali. By the end of 2019, over 200,000 people had been displaced in Mali, and 900 civilians were killed in over 1300 recorded security incidents. Military-led counter-terrorism operations that were deployed to protect civilians largely backfired as they incited revenge attacks, thus adding to the humanitarian crisis. Over the years, these armed groups have expanded, spreading violence throughout Mali and the neighboring countries. The violence in Mali made its way to Burkina Faso in 2018, resulting in close to 500,000 displaced people. For the first time in its history, Burkina Faso experienced

inter-communal and terrorist attacks due to ethnic divisions caused by this regional conflict. In the midst of this crisis, Niger had an influx of people coming in from its southern and western borders. Additionally, an unstable government, banditry, inter-communal clashes, food insecurities, and severe flooding in 2019 resulted in Niger becoming the epicenter of this humanitarian crisis [2]. Those who survived the violence were forced to flee their homes and sought refuge in refugee and internally displaced camp settlements. Despite the urgency of this crisis, the Norwegian Refugee Council in its annual review found the Sahel crisis to be the world's most neglected displacement crisis in 2019 [2]. Moreover, a lack of political will among the governing bodies, drought, hunger, etc., have caused the situation to worsen over time. Adding to this, misplaced media attention, which tends to focus on the *war strategies, fighting between armed groups*, or *political alliances*, was found to be one of the key reasons for the world's most neglected displacement crisis of 2019 [2], and for the top 10 neglected crises in the subsequent years 2020 and 2021 [3,4]. However, the current state-of-the-art research does not sufficiently present curated news media datasets for this region, to study how the media thematically and spatially covers the refugee crisis in West Africa. Furthermore, studies to systemically identify factors that influence media coverage in West Africa are still very nascent.

Therefore, to bridge these gaps in the state-of-the-art, and thereby investigate the media negligence of the crisis in the Sahel region from a spatio-thematic perspective, we conducted a study that examines the spatial foci of online news reporting on the refugee crisis in West Africa and the various themes associated with space in the context of the crisis. This is achieved by curating a dataset for West Africa based on online news articles related to the ongoing crisis, specifically to investigate the spatial foci of news reporting, the regions that are mostly overlooked, and the topics that are covered in the different cities/villages in West Africa. Furthermore, to investigate the factors that influence media coverage in West Africa, this study systemically identifies determinants of news coverage by conducting a linear multiple regression analysis with a combination of political, economical, and social statistical variables from the region. This study is presented in this paper as follows: Section 2 will discuss related works on the coverage and framing of news media during crisis situations, followed by a description of the data and the methodology in Section 3, and the analysis findings in Section 4. The paper will conclude with a discussion of the main findings and a road map for future research in Sections 5 and 6.

## 2. Background & Related Work

According to [5], over 100 million people live outside of their home states, detached from any sense of belonging to a political community, and stripped from access to rights or state protection [6]. Ref. [6] further discusses crises worldwide from a moral angle, how the displaced people are treated from exile to encampment, and the moral obligation for just refugee regimes that fully consider the rights of displaced people before considering the sovereignty of individual host nations. Many facades to the refugee crises around the world have been studied in the literature, ranging from the migration and re-settlement patterns of displaced people (e.g., [7,8]), detection and classification of refugee camp structures (e.g., [9–11]), inter-communal conflicts between refugees and host communities (e.g., [12–15]), and the trauma of displacement (e.g., [16]), to the media coverage and framing of crises around the world (e.g., [17–19]). The literature analysis in this section will focus on how the media coverage and framing of different refugee crises around the world have been studied and relate to the study presented in this article.

The media coverage given to countries across the world differs largely and is grounded by unique structures [20]. Ref. [21] discussed the issue of the volume of Western news coverage of crises, and explained that without a political or an economic angle to report on, the coverage tended to be limited, stating politics and economics to be the biggest determinants of news coverage. Furthermore, she highlighted that selecting stories and how to cover them also largely depend on the practical aspects of news gathering such as access to sources, travel logistics to remote areas, or relying on other media organi-

zations, government agencies, or relief charities to disperse news. Earlier works such as [22–26] have shown that the news coverage received in different countries is governed by factors that are influenced by various national traits (e.g., GNP, population), interaction and relatedness (e.g., trade, import/export volume), or logistical factors related to news gathering (e.g., communication infrastructure, human resources). Ref. [20] performed a critical analysis of the above works and found that most of these findings were based on descriptive statistics that did not help to understand the relationship between the derived factors (independent variables) and news coverage received in a given country (dependent variable). He went on to derive systematized factors that influence news coverage, based on multiple regression, and found *trade volume* and *the presence of international news agencies* to be the most deterministic predictors of news coverage. Ref. [27] assessed the news coverage of the SARS crisis (2002–2004) and its societal impact by analyzing 17 news sites in seven regions, published on 31 March 2003. He found *societal relevancy* and *internet press freedom* to be the determinants of SARS coverage. Ref. [28] compared the news coverage in Belgium before and during the European migrant crisis in 2015 to assess the discourse of media reporting. They analyzed television news items from two broadcasters (one commercial and one public) for the time period 2003–2017. The news coverage in crisis and non-crisis times appears to vary significantly. Similar coverage patterns on immigration were observed among both broadcaster types pre-crisis, and during the crisis, the coverage almost doubled and the commercial broadcaster tended to criminalize immigration and focus predominantly on political actors. Contrastingly, the public broadcaster's focus was broader and covered the point of view of citizens and immigrants during the crisis. As evident from these works, determinants of news coverage vary largely based on the crisis type, the country and the people being covered, and the sampling methods of the data.

How the world events, countries, or its people are portrayed by the news media in every country is inevitably distorted [20]. News framing is the process of selective reporting, where some elements of news are made salient while others are obscured based on the journalists' belief/value systems and external influences [29]. Many works have carried out content analysis tasks on news articles to aid the identification of news frames. Ref. [30] used machine learning approaches to automatically identify dominant news frames in over 10,000 Austrian newspaper articles that covered the arrival of refugees in Europe from 2015 to 2016. Their findings indicate a prominent pattern of framing the security of citizens, the economization of refugees, and, to a lesser extent, the circumstances of the refugees or the humanitarianism during the crisis. Ref. [31] studied 247 news stories related to crises and categorized them according to the type of news frames used (*attribution of responsibility, human interest, conflict, morality, economic*), type of crisis covered, and the level of responsibility. They found that the human interest and morality frames were not frequently used in their sample of news articles, and the articles that did use these frames mostly assigned an individual level of responsibility rather than an organizational level of responsibility. Ref. [32] conducted an automated content analysis of Canadian news media over a 10-year period to identify the discourse of framing of refugees and immigrants and found distinct differences. Immigrants were framed in terms of economic terms, and refugees were framed in terms of security threats, the burden on the social system, and their national origins, with a particularly negative tone to the framing. Their findings further indicate a distinct preference for immigrants over refugees.

Ref. [33] studied the content and the language usage in 1200 news articles on the 2015 refugee crisis, and found three linguistic practices with which the *voice of the refugees is managed* in the news. These are *silencing, collectivization, and de-contextualization*. The narration of the news in Europe was found to be divided between the *us* and *them*, and as little as 16.6% articles included quotes directly from refugees/migrants, and 66% of the sample news articles included quotes from national or EU politicians, which drove the tone of the news. Ref. [34] quantitatively assessed the bias of news framing of the Russian state-owned news agency ITAR-TASS by analyzing the English language content that appeared in over 35,000 news wires on the Ukraine crisis (during 2013–2014). His

findings indicate that most of the news was driven by opinions and politicized agendas, as opposed to the factual reporting of journalists, and was highly biased based on the political relationship between Russia and Ukraine pre- (positive framing of Ukraine) and post- (negative framing of Ukraine) crisis.

In a similar setup, ref. [29] assessed the framing of the Rohingya refugee crisis (2017) through semantic network analysis and text analysis of a total of 747 news stories that appeared in three newspapers from three countries—*The Irrawaddy* (Myanmar), *The New Nation* (Bangladesh), and *The New York Times* (USA)—during a one year period from August 2017 to July 2018. Their findings conclude that *The Irrawaddy*'s news content mostly incorporates pro-government nationalist views whilst making light of the violence against the Rohingyas. *The New Nation* from Bangladesh framed the crisis in terms of the humanitarian aspects, while focusing on their own politicized agendas, and *The New York Times* focused prominently on the frames *surviving horrors*, and *press freedom*, conforming to the patterns of Western media's portrayal of crises in the third-world countries, as observed in [35–38].

Framing biases of news reports, particularly on refugee crises around the world, have been studied in depth in these works, where Eurocentric, Westcentric, elitist, and news framing based on politicized agendas, versus reporting directly from the affected countries, have been discussed at length. However, in crisis situations such as the ongoing plight of refugees in West Africa, in-depth content analyses are lacking, as good data that helps to explore the media coverage or framing in these countries is often scarce both in quantity and quality.

Various datasets have been curated by, for example, [39–44] combining data from the UNHCR repository with data from various NGOs, such as ACLED (https://acleddata.com (accessed on 1 March 2023)) or the World Bank (https://www.worldbank.org/en/home (accessed on 1 March 2023)), and satellite imagery, etc., to study the various aspects of the crisis in West Africa, ranging from refugee movements, violence against refugees, the conflict between refugees and the host population, among others. This paper contributes to the related research by assessing the spatial coverage of the refugee crisis in West Africa as reported on online news media sources, systemic factors that influence this coverage, and a content analysis of the news based on a curated dataset.

## 3. Data and Methodology

On the one hand, a dataset, which we call *The West African Refugees Reporting* (WARRe) was curated to explore the media coverage and the media content on the refugee crisis in West Africa in terms of the spatial foci, the regions neglected in the media, and the reported topics in West Africa. On the other hand, the UNHCR data repository for refugee/IDP locations was used in this study to assess the completeness of the spatial coverage regarding the refugee crisis in West Africa during the considered time period. The following will describe the data sources, the data coding procedure, and the methodology in detail. A summary of the data used in this study is shown in Table 1.

**Table 1.** A summary of the data-sets used.

| | Data Source | | |
|---|---|---|---|
| | **GDELT** | **WARRe** | **UNHCR** |
| **Spatial** | Locations mentioned in the news media in the context of the refugee crisis. These locations are extracted using GDELT's own NLP methods and geocoded | Use GDELT's format of locations, and manually identify the news articles that have more than one location mentioned, code them with parent/child IDs | contains the official records of locations for refugee-settlements |

**Table 1.** *Cont.*

| | Data Source | | |
| --- | --- | --- | --- |
| | **GDELT** | **WARRe** | **UNHCR** |
| **Thematic** | NLP methods are used to automatically classify news articles into a class "Refugees" | Manually examine the news articles and identify the various themes reported on in the context of the "Refugees" | none |

*3.1. Data*

**WARRe data**: The WARRe dataset is primarily based on the freely available data on the GDELT media repository [45]. GDELT stands for *Global data on events, location, and tone*, and is the largest open database on world news to date, which comprises news articles from various sources, such as AfricaNews, Agence France Presse, Associated Press Online, Associated Press Worldstream, BBC monitoring, Christian Science Monitor, Facts on File, Foreign Broadcast Information Service, United Press International, Washington Post, New York Times, Associated Press, Google News, Xinhua, etc. It collects news from every corner of the world at 15-minute intervals, and these news articles are archived and available for public use through a public API for a maximum of 7 days from the time any given news article is first published. The GDELT data repository is sorted by the following categories: *events, counts, quotes, people, organizations, locations, themes, emotions, embedded imagery, and videos*, as well as *embedded social media posts*. GDELT relies on Google's Cloud Natural Language API (https://cloud.google.com/natural-language (accessed on 15 August 2022)) to annotate the online news articles.Various natural language processing techniques for entity recognition and part-of-speech tagging have been used to classify the news sources and extract over 300 categories of events, millions of themes, the associated emotions, and the networks that bind them all together. We were interested in the data that has already been classified under the theme of "Refugees", and started harvesting news articles from the beginning of this study on 12 March 2021 until 15 September 2021, after which the dataset was prepared for analysis.

**UNHCR data**: UNHCR operates its own Geographic Information System (https://www.arcgis.com/home/webmap/viewer.html?webmap=24cad2271eaf4219832bf82da5803193 (accessed on 15 August 2022)), which enables the visualization of geographic data relating to refugees and displaced people. Their underlying data were accessed through a client environment and stored for the comparison and validation of the locations found in the news media related to the refugees. The locations found in this repository are point locations, where larger camps, which should essentially be represented as polygons that contain four geographical coordinate sets, are represented by one geographical coordinate set. These locations represent 30 different categories of settlements, accommodations, or other living arrangements related to the refugee and displaced people in West Africa. A detailed description of these categories can be found in the above link.

**Coding and sorting of WARRe data**: The news articles classified under the theme of "Refugees" in GDELT were downloaded and sorted into chronological order for the time period 12 March–15 September 2021. These articles were then geographically filtered to include only those reporting on West African countries (https://data2.unhcr.org/images/documents/big_f1b633f9a1e0c4c417656631a218ba782703a1c0.jpg (accessed on 15 August 2022), and assigned a unique ID in sequential order. Deepl (https://www.deepl.com/translator (accessed on 15 August 2022)) and Google Translate (https://translate.google.com/ (accessed on 15 August 2022)) services assisted in translating articles and posts written in a language other than English or Spanish. A priori, it is unclear what content-specific thematic categories exist in the news articles on the refugee crisis. Against this background, we had to approach the process manually and exploratively. Therefore, each

news article was examined chronologically to identify the theme(s) reported on. The first time a theme is identified (e.g., *Relocation*), a formal definition of that theme is established (see Table 2), in order to maintain consistency of the classification. As such, when reading the articles, new thematic classes were identified and added to the list along with their definitions (Table 2). Accordingly, 42 relevant thematic classes were identified in 2017 news articles during the considered 6-month time period. If a given article reported on more than one theme (e.g., *Relocation*, *Terrorism*), then all of these thematic classes were assigned to the article. Articles that included no reference to refugees, or were no longer available online were classified as *No relevance* and *No longer available*, respectively. Articles with assigned thematic classes (excluding "No relevance" and "No longer available" categories) were downloaded and saved in HTML format. The final dataset consists of an attribute table containing the GDELT format (including the geographical locations mentioned, URL to the articles, etc.) and the thematic classes assigned to each news article. An excerpt of this dataset is presented in Table 3.

**Table 2.** Identified thematic classes in the WARRe dataset and their descriptions.

| No. | Thematic Class | Thematic Class Description |
|---|---|---|
| 1 | Census | An official count or survey of a population conducted in the refugee camp or host country |
| 2 | Civil war | A violent conflict between organized non-state actors and the state that forces people's displacement |
| 3 | Condition of refugee camp | Infrastructure of the refugee camp/shelter and inhabitants of the refugee camp |
| 4 | Counterterrorism operation | The practice and strategy that government/militia use to combat or prevent terrorism |
| 5 | Development aid | An aid given by governments/agencies to support the economic, environmental, social, and political development of West African countries in terms of refugees/IDPs. Includes articles that talk about people/organization/stakeholders seeking help for refugees |
| 6 | Diseases | A cause of forced migration that deteriorates the health of people and/or animals, e.g., measles, coronavirus |
| 7 | Drought | Deficiency of precipitation over an extended period of time, resulting in a water shortage as a contributing factor for displacement |
| 8 | Elections | Process in which people vote to choose a person or group of people to hold an official position |
| 9 | Expansion of refugee location | Increasing the size of the refugee camp |
| 10 | Expulsion | The action that a government exerts on the inhabitants to abandon a place |
| 11 | Famine | The extreme scarcity of food that drives individuals to look for another place to settle down |
| 12 | Fire | Burns and/or explosions that lead to the displacement of the inhabitants of the area |
| 13 | Flood | An overflow of a large amount of water beyond its normal limits forcing the settlers to leave the territory |
| 14 | Food security | The state of having reliable access to a sufficient quantity of affordable, nutritious food |
| 15 | Gender-based violence | Any act of violence perpetrated against a person's will based on gender norms and unequal power relationships |
| 16 | Herder-farmer conflict | Also called Fulani Herdsmen terrorism, involved disputes over land resources between mostly Muslim Fulani herders and mostly Christian farmers across Nigeria |

**Table 2.** *Cont.*

| No. | Thematic Class | Thematic Class Description |
|---|---|---|
| 17 | Human trafficking | The act of transporting or coercing civilians/refugees/IDPs in order to benefit from their work or service |
| 18 | Independence | Freedom to make laws or decisions without being governed or controlled by another country, organization, etc. |
| 19 | Intercommunal tensions | Arguments or encounters between previous residents and new arrivals or vice versa |
| 20 | Interview | A meeting in which someone answers questions related to refugees/IDPs |
| 21 | Insect plague | When insects or parasites invade an area depleting resources and food |
| 22 | Kidnapping | The action of abducting someone and holding them captive |
| 23 | LGBTQ | Refugee news concerning the LGBTQ community |
| 24 | Landslide | A collapse of a mass of earth or rock from a mountain or cliff that causes forced displacement |
| 25 | Massacre | The indiscriminate and brutal slaughter of many people |
| 26 | Medical aid | An aid given by governments or agencies to support the West African countries in terms of health care |
| 27 | Official visit | A formal visit by a country representative at a refugee camp or shelter |
| 28 | Opinion | An article that expresses the opinion of its editors on refugee-related issues. It also includes topics such as religion, law, and use of natural resources (oil, forest) |
| 29 | Organized crime | Any behavior, activity, or event that is punishable by law, e.g., assault, murder, banditry, armed attacks, and non-state armed groups |
| 30 | Peace talks | A conference or series of discussions aimed at ending hostilities in West Africa |
| 31 | Political statement | A newspaper article describing what politicians/agencies have said about refugees in West Africa |
| 32 | Property damage | A damage or destruction of personal property, e.g., camp facilities, public places, NGO infrastructure |
| 33 | Relocation | As a result of persecution, conflict, generalized violence or human rights violations a person from a country moves or wants to move to any other one, e.g., from a West African country to Europe, from a West African country to another one in Africa, from a country in Africa to West Africa, asylum seekers |
| 34 | Report | An article that gives information about issues, events, or findings in West Africa in the context of refugees |
| 35 | Resettlement | A human movement that is not connected to force, e.g., drought |
| 36 | Returnees | Refugees/IDPs person who escape from a country and return to their previous place of residence |
| 37 | Riots | A violent public disturbance that leads to the movement of people because of the destruction/insecurity generated |
| 38 | Soil degradation | A change in the state of soil quality that causes a decrease in the ecosystem's capacity to provide goods and services and drives forced migration |
| 39 | Stateless | A person who is not considered as a national by any State under the operation of its law |
| 40 | Storms | A violent disturbance of the atmosphere with strong winds and rain that force people to move from their place of residence. It includes rain storms, thunderstorms and wind storms |
| 41 | Terrorism | The use of violence and intimidation, especially against civilians, in the pursuit of political aims as a cause of forced displacement |
| 42 | Water shortage | The lack of fresh water resources to meet the standard water demand in a city/refugee camp |

**Table 3.** An excerpt of the WARRe dataset.

| ID | Location | State | Source URL | Date | Parent_ID | Thematic Classes | News_ Duplicate | Loc_ Duplicate |
|----|----------|-------|------------|------|-----------|------------------|-----------------|----------------|
| 1 | Bamako | Mali | http://s.dlr.de/WCPIt | 12 March 2021 | 1 | Development aid/Gender-based violence/ Relocation | 0 | 0 |
| 2 | Bamako | Mali | http://s.dlr.de/WCPIt | 12 March 2021 | 1 | Development aid/Gender-based violence/ Relocation | 1 | 0 |
| 4 | Kaduna | Nigeria | http://s.dlr.de/fy5F6 | 12 March 2021 | 4 | Organized crime/Kidnapping/Property damage/ Development aid | 0 | 0 |
| 8 | Abidjan | Côte d'Ivoire | http://s.dlr.de/WCPIt | 12 March 2021 | 1 | Development aid/Gender-based violence/ Relocation | 0 | 1 |
| 9 | Abidjan | Côte d'Ivoire | http://s.dlr.de/WCPIt | 12 March 2021 | 1 | Development aid/Gender-based violence/ Relocation | 1 | 1 |
| 10 | Abidjan | Côte d'Ivoire | http://s.dlr.de/fiOQk | 12 March 2021 | 10 | Development aid | 0 | 0 |
| 11 | Abidjan | Côte d'Ivoire | http://s.dlr.de/ELdqe | 12 March 2021 | 11 | Development aid/Condition of refugee camp/ Political statement | 0 | 0 |
| 12 | Djibo | Burkina Faso | http://s.dlr.de/GB64v | 12 March 2021 | 12 | Official visit/Returnees/Development aid/ Relocation | 0 | 0 |
| 13 | Birnin Gwari | Nigeria | http://s.dlr.de/fy5F6 | 12 March 2021 | 4 | Organized crime/Kidnapping/Property damage/ Development aid | 0 | 1 |
| 14 | Abuja | Nigeria | http://s.dlr.de/88EIg | 12 March 2021 | 14 | Interview | 0 | 0 |

**Table 3.** *Cont.*

| ID | Location | State | Source URL | Date | Parent_ID | Thematic Classes | News_ Duplicate | Loc_ Duplicate |
|----|----------|-------|-----------|------|-----------|------------------|-----------------|----------------|
| 15 | Abuja | Nigeria | http://s.dlr.de/X2xOQ | 12 March 2021 | 15 | Interview | 0 | 0 |
| 20 | Lake Chad | Nigeria | http://s.dlr.de/sC640 | 12 March 2021 | 20 | Terrorism | 0 | 0 |
| 21 | Lake Chad | Nigeria | http://s.dlr.de/jnMKE | 12 March 2021 | 21 | Elections/Opinion | 0 | 0 |
| 26 | Tudun Wada | Nigeria | http://s.dlr.de/fy5F6 | 12 March 2021 | 4 | Organized crime/Kidnapping/Property damage/ Development aid | 0 | 1 |
| 27 | Birnin Yero | Nigeria | http://s.dlr.de/fy5F6 | 12 March 2021 | 4 | Organized crime/Kidnapping/Property damage/ Development aid | 0 | 1 |
| 28 | Igabi | Nigeria | http://s.dlr.de/fy5F6 | 12 March 2021 | 4 | Organized crime/Kidnapping/Property damage/ Development aid | 0 | 1 |
| 29 | Koulouba | Mali | http://s.dlr.de/WCPIt | 12 March 2021 | 1 | Development aid/Gender-based violence/ Relocation | 1 | 1 |
| 30 | Sanmatenga | Burkina Faso | http://s.dlr.de/GB64v | 12 March 2021 | 12 | Official visit/Returnees/Development aid/ Relocation | 0 | 1 |
| 32 | Sansani | Niger | http://s.dlr.de/FSzMT | 12 March 2021 | 32 | Kidnapping/Terrorism | 0 | 0 |
| 33 | Souleymane | Mali | http://s.dlr.de/GB64v | 12 March 2021 | 12 | Official visit/Returnees/Development aid/ Relocation | 0 | 1 |

Duplicate articles were identified based on the article's content and the place of publication. Due to the fact that articles or web posts can be published, updated, or republished on different media platforms at the same time or with a time lag, there may be several articles with identical titles and content. In order to make these republished articles recognizable, these duplicates were marked in a designated attribute named "news duplicate". These duplicates received their unique ID as well as a database identifier (ID) of the first occurring entry of the article (i.e., the "parent ID"). Furthermore, an article can reference not only one but several different geo-referenced locations. Thus, multiple features may exist in the data set for each article. These duplicate features have been flagged in the corresponding attribute "location duplicate" and receive the database identifier (ID) of the first occurring entry of the article (i.e., the "parent ID") (see Table 3). This can result in different combinations of "news duplicates" and "location duplicates". This procedure allows different options for filtering the data set. At the end of the data coding for each month, the following checks were performed: data cleaning, consistency, completeness, and metadata availability.

**Usage notes**: The definition of the thematic classes and the classification of the articles in this dataset are the results of an extensive manual process. The goal was to represent the various content aspects of the articles with the highest degree of thematic accuracy, which naturally leads to a high granularity of categories. The annotation of the articles was performed to the best of our knowledge by very few experts to maintain consistency. Nevertheless, the subjective interpretation of the thematic classes is inherently biased. These thematic classes can be used as is or grouped into broader categories, depending on the purpose of the investigation. It should be noted that due to a technical inconsistency at the beginning of the data harvesting, no data are available for 13 March 2021. Furthermore, the date of each dataset entry reflects a combination of the date when GDELT captured the article and the time zone we used for scheduling our downloads from GDELT. For example, an entry labeled "15 March 2021" was captured by GDELT at some time between 13 March 2021, 12 a.m., Central European Time and 14 March 2021, 12 a.m., Central European Time.

**The WARRe data**: Table 2 lists all 42 thematic classes along with their definitions, as identified in the news articles throughout the annotation process for the time period 12 March to 15 September 2021. These definitions were consistently applied during the annotation process of the news articles. Table 3 is an excerpt of the WARRe dataset (due to space limitations, only the most important attributes of the table are presented here. The complete table will be presented as supplementary material after the review process or on request). The first column (attribute) **ID** is the unique identifier of the individual rows (features) corresponding to the news articles. **Location** and **State** are the city and country in which the city is situated, corresponding to the location(s) mentioned in the news article. The **Source URL** is the web address of the news articles (the lifespan of the URL may differ according to its host). **Date** is the downloaded date of the news article, and **Parent_ID** is the ID of the first occurring entry of location and news duplicates. In news articles where more than one location is mentioned, the entry is duplicated for each subsequent location, and identified by its **Parent_ID**. These duplicates are indicated in the table under the attributes **news_duplicate** and **loc_duplicate**. The features under these duplicates are indicated with either a "0" for false (when they are not duplicates) or "1" for true values (when they are duplicates).

*3.2. Methodology*

This section describes our analysis strategy for exploring the spatial coverage of news articles across West Africa in the context of the refugee crisis, and for identifying factors that influence news coverage.

**News coverage**: News coverage is the media attention given to the countries in West Africa with respect to the topic of refugees, and it is assessed by the total number of times a country is mentioned in the news (*n* in Table 4) throughout the 6 months considered in this study. It is represented as a proportion of the total number of times that

all countries in West Africa are mentioned in the news articles in the study. These counts are visualized as a heatmap table to obtain a visual overview using the JMP statistical software (https://www.jmp.com (accessed on 15 August 2022)) (Figure 1). To understand the news coverage with respect to the refugee/displacement crisis in West Africa, a comparative analysis was performed between the WARRe data of news media coverage and the UNHCR data of all refugee/ IDP locations in West Africa. To achieve this, and to aid a visual comparison, all the geographic locations found in the annotated WARRe data and in the UNHCR data repository were mapped with contrasting symbols in two separate feature layers in a GIS environment (Figure 2). As a next step, a spatial proximity analysis was conducted with both datasets, to identify how many locations found in the media overlap with, or are in close proximity to the locations in the UNHCR repository. To achieve this, a buffer zone is created to separate the locations in the WARRe dataset that are at, or at a 2 km distance to the nearest UNHCR locations (assuming that a 2 km radius accounts for larger camps in the UNHCR repository that are represented as points instead of polygons). These intersecting points are represented in a separate layer, and given a different symbol in the GIS environment (Figure 2). The JMP statistical software is used to count the number of unique locations present in all three layers.

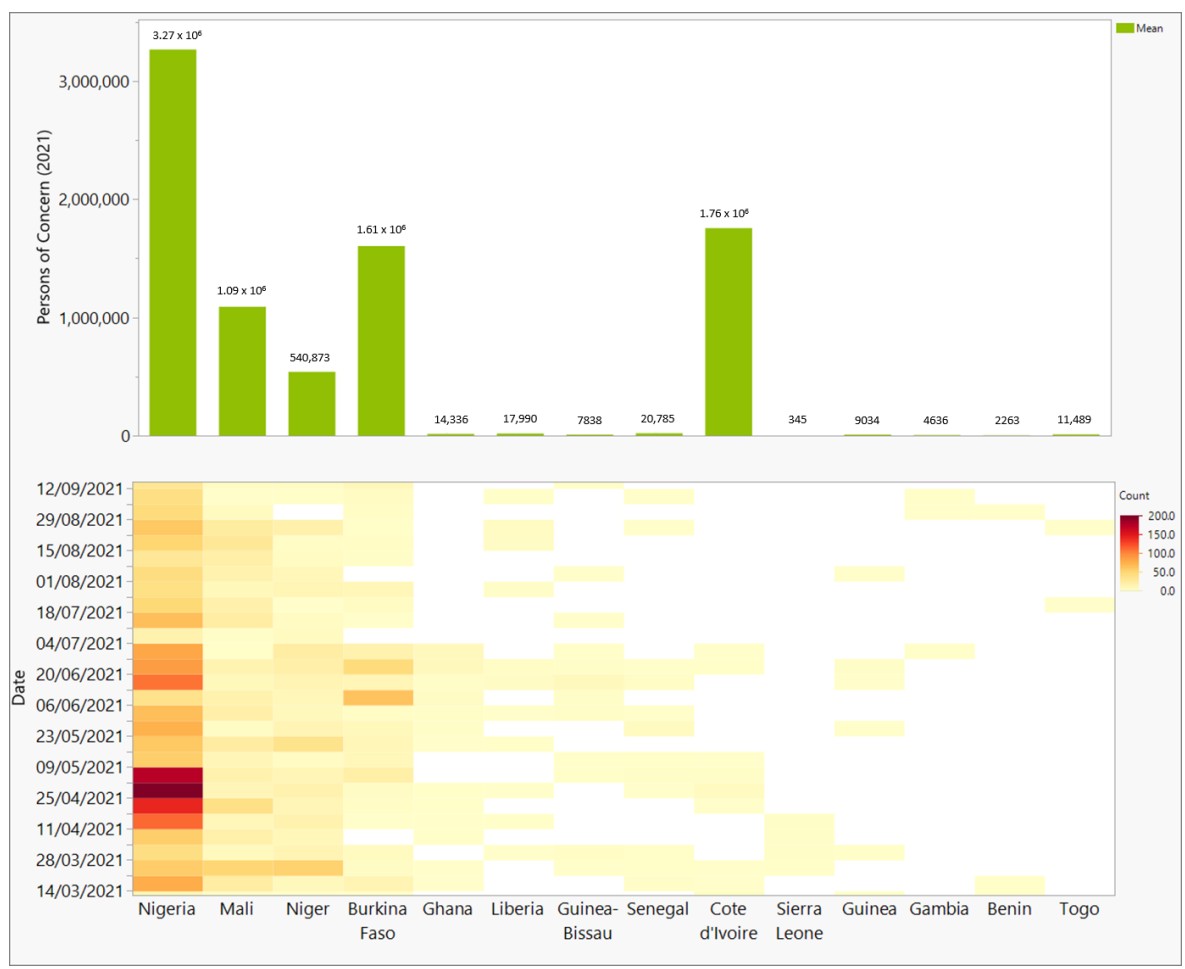

**Figure 1.** The media coverage given to each country over the 6 months considered in this study (bottom) in comparison with the total number of persons of concern in each country (up).

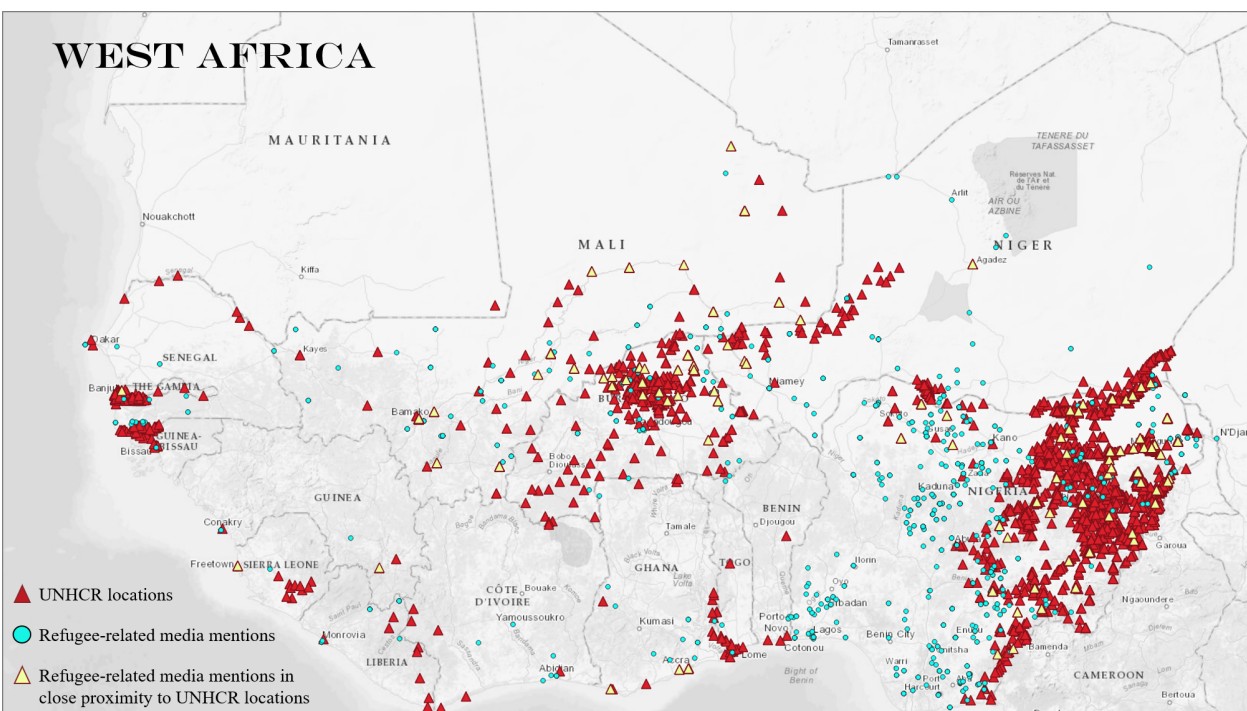

**Figure 2.** The locations recorded in the UNHCR database (red), the locations mentioned in the news media referring to the refugee crisis (blue), and media mentions in close proximity to the locations recorded in the UNHCR during the time period 12 March–15 September 2021.

**Content analysis**: The aim of this step is to systematically study the content found in the news reports. The thematic annotations in the WARRe dataset help to identify the topics associated with locations in the West African region in the context of the refugee crisis, and how these change over time, all through the eyes of news reporting. Each of the 42 thematic classes found in this study is visualized in terms of the total number of news articles that represent these thematic topics, and in terms of their spatial coverage. The number of news articles corresponding to a given thematic class is aggregated over the 6-month period considered in this study and is represented as a proportion of the total number of news articles and visualized as a Treemap (Figure 3). This gives an overview of the most and least reported topics regarding the refugee crisis in West Africa during the considered time frame. The JMP software is used to do the aggregates and to visualize these thematic categories. To identify the location hotspots corresponding to the thematic classes, a Kernel Density Estimation (KDE) was performed on the locations mentioned under each thematic class, for the 6 months considered. These hotspots of locations are visualized as heatmaps in the GIS environment (Figures 4–7).

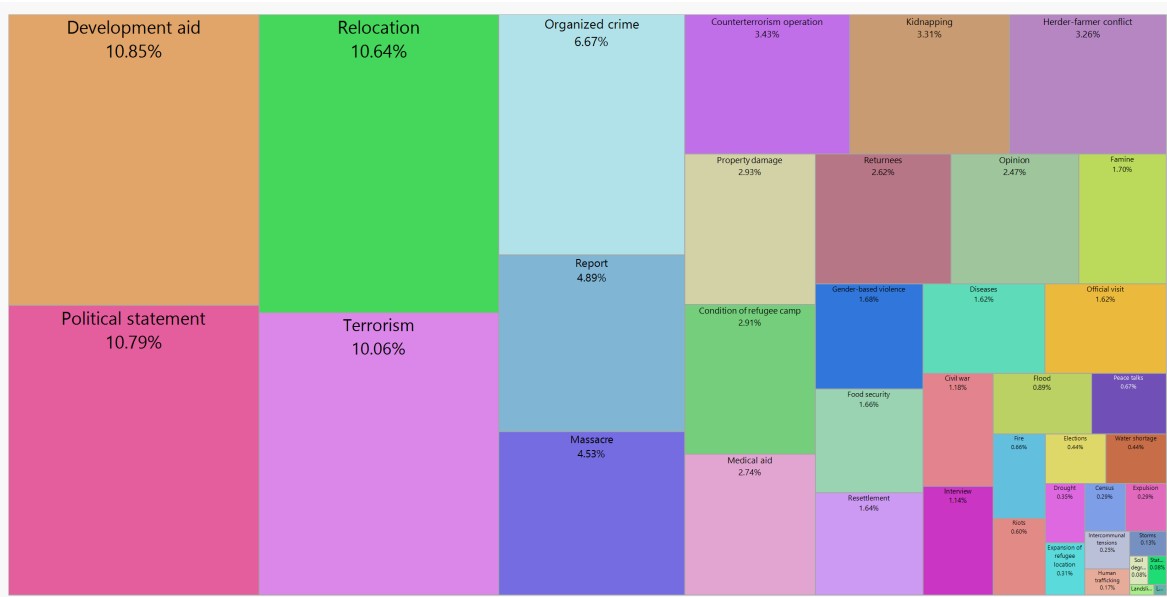

**Figure 3.** An excerpt of the thematic classes found in the news articles aggregated over the 6-month period considered in the study and sorted.

**Systemic determinants**: In our investigation, we aim to explore and identify what factors influence the news media to report on a country in West Africa regarding the refugee crisis, and thereby sought clarification for the disproportions in the spatial coverage of news reporting across this region. This is achieved by deriving *systemic determinants* of news coverage through a linear multiple regression analysis, similar to the work of [20]. Systemic determinants are defined by [20] as *the distinctive traits of individual nations, as well as the magnitude of the interaction between any two nations in the context of the global system.* As the dependent variable in the linear multiple regression analysis, we choose the total number of times a country is mentioned in the news (*n* in Table 4), in the 6-month period considered in this study. To test the co-linearity with news coverage, 13 independent variables that are reflective of various national traits of countries are chosen based on previous similar studies for systemic determinants of news coverage for various regions around the world (see [20,21,23–26]), and based on the availability of data for these variables that describe various traits of the countries in West Africa. These are: *population of the country* (https://www.worldatlas.com/ (accessed on 1 April 2022)), *geographic size* (https://www.worldatlas.com/ (accessed on 1 April 2022)), *annual GDP* (https://countryeconomy.com/countries/groups/economic-community-west-african-states (accessed on 1 April 2022)), *GDP per capita* (https://countryeconomy.com/countries/groups/economic-community-west-african-states (accessed on 1 April 2022)), *refugee population* (https://data.worldbank.org/indicator/SM.POP.REFG?locations (accessed on 1 April 2022)), *Fulani ethnic population in the region* (https://www.hauniversity.org/en/Peul-Fula.shtml (accessed on 1 April 2022)), and world governance indices for the year 2021 for: *voice and accountability, political stability and no violence, government effectiveness, control of corruption, regulatory quality, rule of law.* These indices of governance are the results of The Worldwide Governance Indicators project [46] conducted by the *Knowledge for Change* research program at the World Bank https://www.worldbank.org/en/programs/knowledge-for-change (accessed on 1 April 2022)). These indices are derived based on the combined views of enterprises, citizens, and expert survey respondents every year. A description of the methodology used to derive these aggregate indices and the list of data sources used can be found in [47]. A description of these variables (except for the *population*, all other variables represent the statistics from the year 2021) are in Table 5.

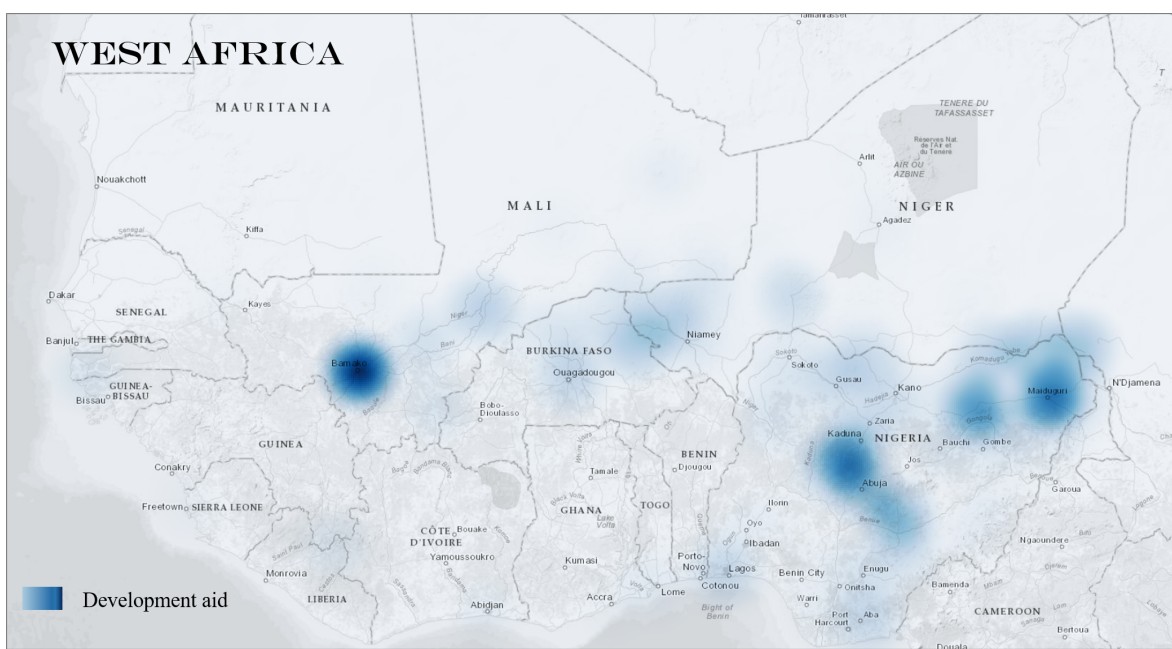

**Figure 4.** The hotspots of locations mentioned in articles that are mostly concerned with the theme of *Development aid*.

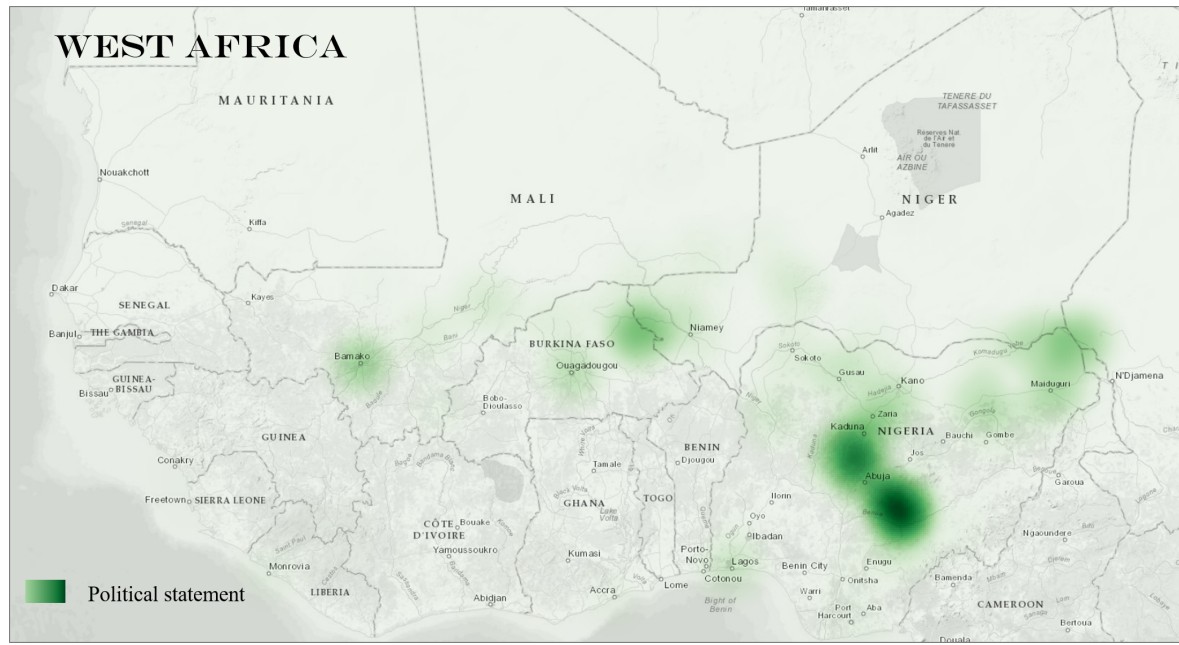

**Figure 5.** The hotspots of locations mentioned in articles that are mostly concerned with the theme of *Political statement*.

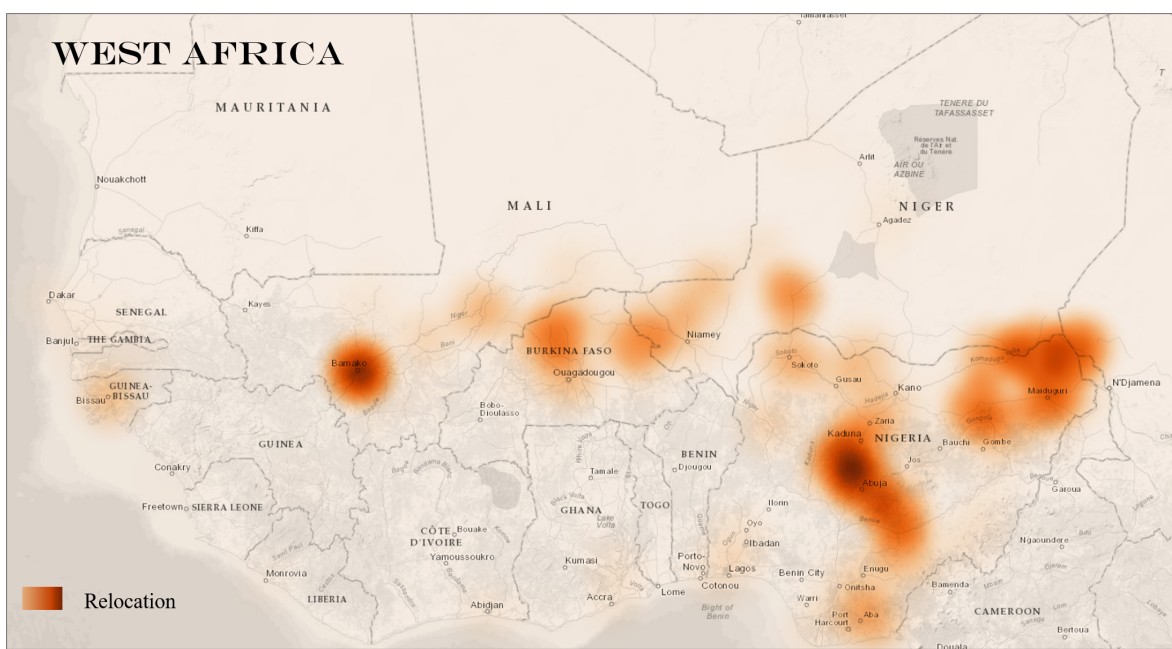

**Figure 6.** The hotspots of locations mentioned in articles that are mostly concerned with the theme of *Relocation*.

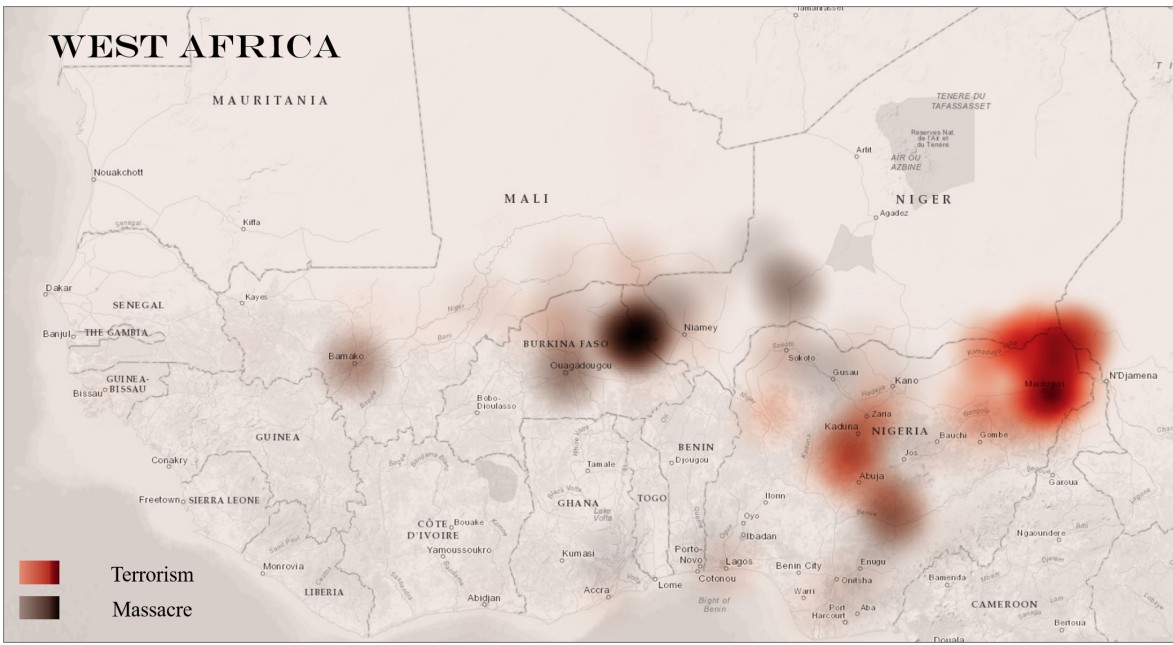

**Figure 7.** The hotspots of locations mentioned in articles that are mostly concerned with the themes of *Massacre* (grey) and *Terrorism* (red).

**Table 4.** Media coverage received by West African countries regarding the refugee crisis.

| Country | *n* | *n*/Total | % |
|---|---|---|---|
| Nigeria | 1974 | 0.60128 | 60.13 |
| Mali | 468 | 0.14255 | 14.26 |
| Niger | 366 | 0.11148 | 11.15 |
| Burkina Faso | 290 | 0.08833 | 8.83 |
| Ghana | 42 | 0.01279 | 1.28 |
| Liberia | 32 | 0.00975 | 0.97 |
| Guinea-Bissau | 31 | 0.00944 | 0.94 |
| Senegal | 28 | 0.00853 | 0.85 |
| Cote d'Ivoire | 21 | 0.00640 | 0.64 |
| Sierra Leone | 11 | 0.00335 | 0.34 |
| Guinea | 9 | 0.00274 | 0.27 |
| Gambia | 5 | 0.00152 | 0.15 |
| Benin | 4 | 0.00122 | 0.12 |
| Togo | 2 | 0.00061 | 0.06 |

By following a backward selection approach, our first step was to perform a univariate test and determine the covariates that correlate with *n*- the number of times a country is mentioned in the news over the 6-month period. The correlation analysis of the variables in Table 5 and the measure *n* is to see if they have a linear relationship. The non-linear variables are removed, and the linear covariates that have a coefficient higher than 0.2 (positive correlations) and lower than $-0.2$ (negative correlations) are used in a multiple regression analysis to further investigate the systemic determinants.

The next step is to run multiple regression. Every chosen covariate has a hypothesis test attached to it. Each hypothesis test has a null hypothesis—$H_0$, and an alternative hypothesis—$H_a$. Every covariate has the following $H_0$: *this covariate is not significant to the regression model* and $H_a$: *this covariate is significant to the regression model*. Our goal is to reject the $H_0$. The threshold for choosing covariates for the multiple regression is the alpha value $p = 0.05$. As the covariates were selected based on a backward selection, we needed to run the model several times until the probability is greater than the absolute value (i.e., *p*-value) of a *t*-test. Because of the co-linearity of the covariates (different variables have different effects on the model), we cannot simply remove all covariates that have a $p > 0.05$. Therefore, by accounting for co-linearity, we keep removing the covariates, going from the highest *p*-values (which fails to reject the null hypothesis), until all covariates that are left are the ones below the 0.05 threshold. In parallel, the adjusted R Squared value had to be observed. The adjusted R Squared value tests the precision of the model when additional variables are added to the linear multiple regression model. Whilst fitting our regression model, the adjusted R Square value was maintained, not going low while removing the insignificant covariates. Once the linear multiple regression model was established, three further tests were performed before finalizing the model. These were the residual test, equal spread condition test, and F-test.

**Table 5.** Description of the variables considered in the multiple regression model.

| Variable | Description |
|---|---|
| Population (2020) | The number of inhabitants of a country in a given year |
| Geographic size (km²) | The total area of land governed by a country |
| Annual GDP ($ - Mil.) | The total market value of all final goods and services produced within a country in a given year, is known as the annual Gross Domestic Product |
| GDP per Capita ($) | The sum of gross value added by all resident producers in the economy, plus any product taxes not included in the valuation of output, divided by mid-year population |
| Refugee population | Persons who are outside their country of origin for reasons of feared persecution, conflict, generalized violence, or other circumstances that have seriously disturbed public order, and as a result, require international protection |
| Fulani Population | The number of inhabitants that belong to one of the largest ethnic groups in West Africa- the Fula ethnic group. |
| International migrant stock | The total number of international migrants present in a given country at a particular point in time |
| Political stability and the absence of violence/ terrorism | Measures perceptions of the likelihood of political instability and/or politically-motivated violence, including terrorism. |
| Control of corruption | Reflects perceptions of the extent to which public power is exercised for private gain, including both petty and grand forms of corruption, as well as "capture" of the state by elites and private interests. |
| Regulatory quality | Reflects perceptions of the ability of the government to formulate and implement sound policies and regulations that permit and promote private sector development. |
| Voice and accountability | Reflects perceptions of the extent to which a country's citizens are able to participate in selecting their government, as well as freedom of expression, freedom of association, and a free media. |
| Government effectiveness | Reflects perceptions of the quality of public services, the quality of the civil service and the degree of its independence from political pressures, the quality of policy formulation and implementation, and the credibility of the government's commitment to such policies. |
| Rule of law | Reflects perceptions of the extent to which agents have confidence in and abide by the rules of society, and in particular the quality of contract enforcement, property rights, the police, and the courts, as well as the likelihood of crime and violence. |

## 4. Findings

Following the methodology in Section 3.2, we have derived findings for spatial and thematic news coverage regarding the refugee crisis in West Africa, and their systemic determinants. These findings are elaborated on in the following sections.

### 4.1. News Coverage of the Refugee Crisis in West Africa in Spatial Terms

Through the manual inspection of the 2017 news articles for the considered 6-month time period, we have found 14 countries, and 505 unique locations (i.e., villages or towns) corresponding to these countries mentioned in the news articles that reported on the refugee crisis in West Africa. Table 4 shows the media coverage received by each country in West Africa, in terms of the total number of times each country was mentioned. Figure 1 draws a comparison between the media coverage received in each country throughout

the considered time period and the total number of *persons of concern* [48] in each country, as recorded in the UNHCR for the year 2021 (https://data2.unhcr.org/en/documents/download/91265 (accessed on 1 April 2022)).

As it becomes evident from this analysis, Nigeria is prominent in the news and is mentioned almost every day as associated with the refugee crisis (see Figure 1). Over 60% of all location mentions in the news articles account for Nigeria. Considering Nigeria's 3.2 million persons of concern according to the UNHCR (https://data2.unhcr.org/en/documents/download/91265 (accessed on 15 August 2022))—the highest number in the region, the high coverage seems only logical. However, in comparison, other countries in West Africa, such as Cote d'Ivoire, where over 1.7 million people are recognized as persons of concern (https://data2.unhcr.org/en/documents/details/91265 (accessed on 15 August 2022)), are lurking in the shadows of the region's economic and demographic giant, with only 0.6% of all location mentions in the news articles accounting for Cote d'Ivoire in the 6-month time period.

Figure 2 shows a comparison of the locations found in the news articles with respect to the refugee crisis, with the locations of the refugee and displaced people recorded in the UNHCR repository. The UNHCR locations are shown in red, the locations mentioned in the media are shown in blue, and the media locations that are at or within a 2 km radius of one or more UNHCR locations are shown in yellow. While a total of 3116 unique locations were recorded in the UNHCR database for the West African region (Figure 2—red), only 99 locations (Figure 2—yellow) found in the news media overlap with or are within a 2 km radius to the UNHCR locations. For example, 96.8% of all refugee-related locations recorded by the UNHCR in this region, do not appear in the media reports at all, during the time period considered in this study. Contrarily, 80.4% of locations mentioned in the news media with respect to the refugees (i.e., 406 out of the total of 505 unique locations found in the media) are not in close proximity to the refugee-related locations recorded in the UNHCR repository. While flight, displacement, forced migration, etc., are ongoing humanitarian crises throughout the region, 3017 individual refugee-related locations in West Africa are not reported on and are neglected according to this analysis.

### 4.2. News Coverage of the Refugee Crisis in West Africa with Respect to Content

Figure 3 shows all the thematic classes found in the news articles aggregated over the observed 6-month time period. Accordingly, *Development aid* (in 10.85% of total news articles), *Political statement* (10.79%), and *Relocation* (10.64%) are the most frequently identified thematic classes in the news articles, followed by *Terrorism* (10.06%), *Organized crime* (6.67%), and *Reports* (4.89%).

Figures 4–6 show the location hotspots that correspond with the most frequently identified thematic classes. Evidently, news reports on *Development aid* mostly mention Bamako in Mali, news reports on *Political statement* mostly mention Benue and Abuja in Nigeria, and news reports on *Relocation* mostly mention Bamako and Abuja locations.

With the help of the thematically annotated classes *Massacre* and *Terrorism*, we could identify location hotspots that correspond to news reports about mass killings and terrorism during the considered time period. An overlay of heatmaps corresponding to these thematic classes is presented in Figure 7. Regions reported as affected by mass killings, i.e., Burkina Faso, Mali, and Niger, are clearly visible in grey, while red hotspots show that north-eastern regions of Nigeria and its bordering regions to Niger and Chad are prominently covered in the news while reporting on Terrorism. The difference in the portrayal of the conflict in terms of massacres taking place in Burkina Faso versus terrorism in Nigeria cannot be overlooked.

It is also evident from this analysis that most other pressing issues related to the refugee crisis in West Africa, such as famine, access to clean water, or sexual violence in camps, are overshadowed by topics such as development aid, political statements, or terrorism. According to a study by Cadre Harmonisé (https://www.oxfam.org/en/press-releases/west-africa-faces-its-worst-food-crisis-ten-years-over-27-million-people-already (accessed on

1 November 2022)), 27 million people in West Africa were suffering from hunger as of March 2022, and this number was expected to rise up to 38 million by June 2022. However, the topic of famine is reported in only 1.7% of the total 2017 news articles that were considered in this study. Ref. [49] revealed in a case study report for the year 2021 that droughts in West Africa are causing production losses, rising food prices, increasing hunger and malnutrition, and causing herder–farmer conflicts. However, only 0.35% of news articles report on drought, and 0.44% on water shortages and their effects on migration, famine, etc., during the 6 months considered in this study. Furthermore, in an assessment of trafficking risks in internally displaced persons by the UNHCR [50], individual camps across eastern Nigeria showed significant rates of sexual exploitation of camp inhabitants. Often, women under the age of 20 are exploited while they are en route to gather water or firewood. A total of 1.7% of news articles reported on gender-based violence, and 0.2% reported on related human trafficking activities for the whole of West Africa during these 6 months. These are a few examples of what is not covered in depth by the world media in the context of the refugee crisis in West Africa, and as the other thematic classes show in Figure 3, there are many more topics that have received only a little coverage by the news media.

### 4.3. Systemic Determinants of News Coverage for the Refugee Crisis in West Africa

In the following, we aim to identify systemic determinants of news coverage. The summary statistics of the linear covariates chosen through the backward selection approach are shown in Table 6, and the $p$ values and the estimates of the variables that are used for the fitting of the multiple regression model are shown in Table 7 (the first iteration of the model) and Table 8 (the second iteration of the model). Accounting for co-linearity, covariates that fail to reject the null hypothesis are removed. As an exception, the *Refugee population* covariate with a $p$ value of 0.0547 was left in the model (see Table 8), as removing it caused the adjusted R Squared value to lower.

Based on the final results, the linear multiple regression model with systemic determinants that predicts the news coverage is as follows:

$$
\begin{aligned}
Newscoverage(Y) \ = \ & -3 \times 10^{-5}(Population) + 0.02(AnnualGDP) - 0.7(GDPperCapita) &&(1)\\
& +0.001(Refugeepopulation) - 188.5(Politicalstability) &&(2)\\
& -339.6(Controlofcorruption) + 964.2(Regulatoryquality) &&(3)\\
& +0.00002(Fulanipopulation) + 1225.2 &&(4)
\end{aligned}
$$

**Table 6.** Covariates chosen through the backward selection for the linear multiple regression.

| Covariate | Mean | Standard-Deviation | Correlation-Coefficient | $p$ |
|---|---|---|---|---|
| Population | 28,313,158 | 51,982,771 | 0.96 | *<0.001* |
| Geographic size | 364,905.4 | 438,759.7 | 0.59 | *0.0246* |
| Annual GDP ($, Mil.) | 48,791.93 | 111,470.7 | 0.94 | *<0.001* |
| GDP per Capita ($) | 1180.93 | 619.213 | 0.32 | *0.2682* |
| Refugee population | 33,265.86 | 66,019.87 | 0.37 | *0.1923* |
| Fula population | 2,476,282 | 3,729,209 | 0.87 | *0.0002* |
| Political stability | −0.796 | 0.769 | −0.61 | *0.0199* |
| Control of corruption | −0.564 | 0.424 | −0.36 | *0.1992* |
| Regulatory quality | −0.63 | 0.346 | −0.26 | *0.3717* |
| Persons of Concern | 598,904.8 | 1,025,295 | 0.81 | *0.0009* |

Italic: statistically highly significant.

**Table 7.** Fitting the multiple regression model with a backward selection of covariates—the first model run.

| First Model Run | | | | |
|---|---|---|---|---|
| **Summary of Fit** | | | | |
| RSquare | 0.99966 | | | |
| RSquare Adj | 0.998131 | | | |
| Root Mean Square Error | 24.24293 | | | |
| Mean of Response | 270.75 | | | |
| Observations (or Sum Wgts) | 12 | | | |
| **Analysis of Variance** | | | | |
| **Source** | **DF** | **Sum of Squares** | **Mean Square** | **F Ratio** |
| Model | 9 | 3,457,106.8 | 384,123 | 653.582 |
| Error | 2 | 1175.4 | 588 | **Prob > F** |
| C. Total | 11 | 3,458,282.3 | | 0.0015 |
| **Parameter Estimates** | | | | |
| **Term** | **Estimate** | **Std Error** | **t Ratio** | **Prob > t** |
| Intercept | 1274.7912 | 227.0906 | 5.61 | 0.0303 |
| Population | −0.00003256 | 0.000009167 | −3.55 | 0.0709 |
| Geographic size (Sq.Km.) | −0.000021 | 0.00006711 | −0.31 | 0.784 |
| Annual GDP ($,Mil.) | 0.0205875 | 0.004289 | 4.8 | 0.0408 |
| GDP per Capita ($) | −0.749812 | 0.109157 | −6.87 | 0.0205 |
| Refugee population | 0.0007415 | 0.000359 | 2.07 | 0.1749 |
| Index: Political Stability- No violence | −194.5868 | 37.60319 | −5.17 | 0.0354 |
| Index: Control of corruption | −363.6182 | 105.2281 | −3.46 | 0.0745 |
| Index: Regulatory quality | 1014.8891 | 211.6453 | 4.8 | 0.0408 |
| Fula population | 0.000023268 | 0.00007683 | 3.03 | 0.0939 |

The residual test determines the expected vs. observed values. As can be seen from the graph in Figure 8a, our regression model is good at predicting news coverage below 500 country mentions in a 6-month period. When the residuals are graphed by themselves, as can be seen from the histogram in Figure 8b, it suggests that the residuals are normally distributed with one extreme outlier. As the relationship is approximately linear, we proceed with the assumption that error terms are normally distributed, thereby satisfying the equal spread condition.

The F-test determines if the established multiple regression model is better or worse than using the mean to predict the future values of news coverage. The $H_0$ of the F-test is that *the regression equation is not better than using the mean*. The $H_a$ is that *the regression equation is better than using the mean*. With a $p < 0.0001$, we reject the $H_0$. Therefore, by using the established regression model, we can determine with over 99% certainty news coverage for the countries in West Africa for a given 6-month period based on the following factors: countries' *GDP per Capita*, *Regulatory quality*, *Political stability*, *Annual GDP*, *Control of corruption*, *Population*, *Fulani population*, and the *Refugee population*. These covariates are described in Table 5. These findings are in resonance with most of the related works in Section 2, where the politics and economics of countries were found to be the most deterministic factors of news coverage during crisis situations in other regions of the world.

**Table 8.** Fitting the multiple regression model with a backward selection of covariates—the second model run, after removing the covariate *Geographic size*.

| Second Model Run | | | | |
|---|---|---|---|---|
| **Summary of Fit** | | | | |
| RSquare | 0.999643 | | | |
| RSquare Adj | 0.998693 | | | |
| Root Mean Square Error | 20.27289 | | | |
| Mean of Response | 270.75 | | | |
| Observations (or Sum Wgts) | 12 | | | |
| **Analysis of Variance** | | | | |
| **Source** | **DF** | **Sum of Squares** | **Mean Square** | **F Ratio** |
| Model | 8 | 3,457,049.3 | 432,131 | 1051.44 |
| Error | 3 | 1233 | 411 | **Prob > F** |
| C. Total | 11 | 3,458,282.3 | | <0.0001 |
| **Parameter Estimates** | | | | |
| **Term** | **Estimate** | **Std Error** | **t Ratio** | **Prob > t** |
| Intercept | 1225.2212 | 136.0482 | 9.01 | 0.0029 |
| Population | −0.00003144 | 0.000007057 | −4.46 | 0.021 |
| Annual GDP ($,Mil.) | 0.0200171 | 0.003246 | 6.17 | 0.0086 |
| GDP per Capita ($) | −0.726707 | 0.067221 | −10.81 | 0.0017 |
| Refugee population | 0.0006636 | 0.000216 | 3.07 | 0.0547 |
| Index: Political Stability- No violence | −188.4586 | 26.84245 | −7.02 | 0.0059 |
| Index: Control of corruption | −339.616 | 60.23026 | −5.64 | 0.011 |
| Index: Regulatory quality | 964.23936 | 114.006 | 8.46 | 0.0035 |
| Fula population | 0.0000226 | 0.000006173 | 3.66 | 0.0352 |

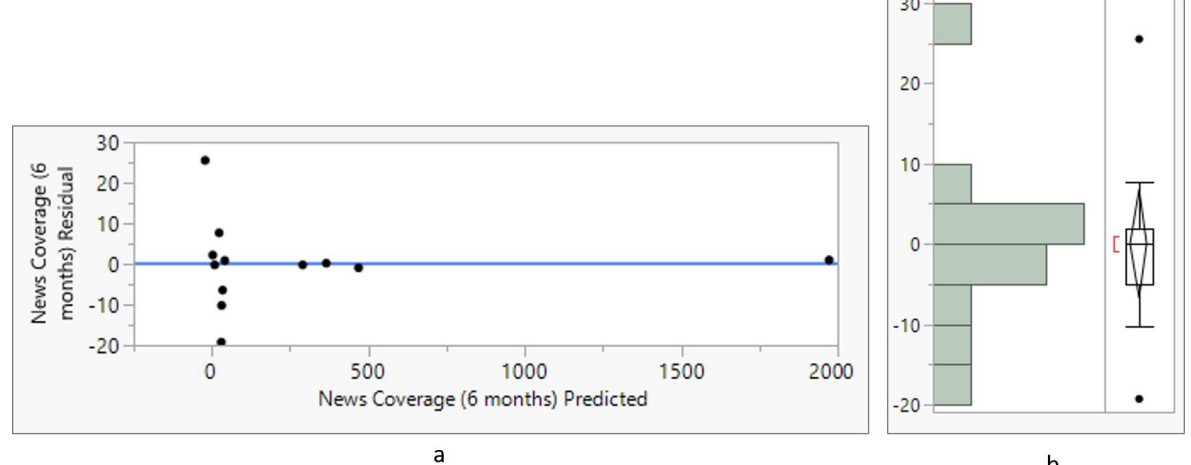

a

b

**Figure 8.** (**a**)—Residual and (**b**)—the equal spread condition tests

## 5. Discussion and Future Research Directions

Within the 6 months of news coverage related to refugees in West Africa considered in this study, we found that **96.8%** of refugee-related UNHCR locations in the West African

region are **not** mentioned in the news media at all, and the majority of the news articles that did cover the crisis reported mostly on Nigeria, the region's "demographic and economic giant" [51]. In addition to the middle belt and around the Niger Delta in Nigeria, central Mali, northern Burkina Faso, and Lake Chad form the largest epicenters of violence, according to a study by Trémolières et al. [1]. However, some of these regions are not well represented in the news articles published during the time period considered in this study. The most widely discussed topics such as *Political statement* or *Relocation* are mostly linked with locations in Nigeria. In their official reports covering the same time frame, the Norwegian Refugee Council, the UNHCR, and ReliefWeb listed emergency events happening across the region with hundreds of civilians dying due to terrorism and government-run anti-terrorism operations. In their accounts, Burkina Faso even reached its highest number of internally displaced people in the region (https://data2.unhcr.org/en/situations/sahelcrisis (accessed on 1 November 2021)) in this time frame. However, while some of these events are reported, they are largely overshadowed by the bulk of reports on Nigeria. Some significant events (by account of death toll) that occurred during the considered time period but have received little to no coverage are the Solhan attack on 5 June 2021 (https://www.nrc.no/news/2021/june/statement-on-the-solhan-attack/ (accessed on 1 November 2021)), repeated attacks on three villages in Niger that killed 137 people, including refugees https://news.un.org/en/story/2021/04/1088872 (accessed on 1 November 2021), organized political violence in Burkina Faso, Niger, and Mali, for which 70% of the fatalities were estimated to be civilians (https://reliefweb.int/report/mali/sahel-2021-communal-wars-broken-ceasefires-and-shifting-frontlines (accessed on 1 November 2021)), or other major incidents (https://m.reliefweb.int/report/3733886/burkina-faso/burkina-faso-violence-against-civilians-dg-echo-ingos-echo-daily-flash-28-april?lang=fr (accessed on 1 November 2021)) (https://www.voanews.com/a/africa_east-burkina-faso-several-dozen-dead-attack/6205402.html (accessed on 1 November 2021)) in Burkina Faso.

The deterministic factors for news coverage found in Section 4.3 explain the biased news reporting towards Nigeria based on factors such as the GDP per Capita, or the political stability in the country, in comparison to the other countries in the region. These findings are in resonance with studies such as [20] or [25], which found the economic and political stability of a country (among others) to be significant determinants of news coverage. This is possibly because economic or political stability allows accessibility, infrastructure, or freedom to report on news stories. Future research should reveal why such factors influence news coverage, and explain how the severity of a crisis plays a role. Several explanations can also be linked to the lack of coverage in other countries, including GDELT's algorithmic biases in the data harvest, and political/cultural news reporting biases present in the region [51]. NRC [2] found disproportions between media attention and the size of the crisis, and they found Burkina Faso in third place and Nigeria in eighth place on a list of the world's most neglected crises, with a lack of media attention being a deciding factor. In addition, Fengler et al. [52] found that an absence of a shared ethnic background between the local communities and the journalists is seen as an obstacle to covering events happening within those communities. Furthermore, a lack of infrastructure and access to remotely located refugee and displaced communities make it difficult for journalists to cover events unfolding in these locations. Camps in which famine, disasters, or starvation have become normal, and which are, therefore, not considered newsworthy, do not draw the attention of the journalists [52]. Another factor influencing the coverage may be linked to that of the readership [53]. The journalists will tailor their press coverage to what the reader wants to hear—for some outlets that means, e.g., that sensational news, dramatic and violent incidents are rather reported than continued starvation in a camp, or the readers are of a certain ethnic background and religion, and will show more sympathy for refugees of the same background and with the same devotion. Journalists may also want to write about what scares or shocks their readers the most. These speculations can be tested in future research with the availability of data.

Another interesting finding is how incidents happening across this region are framed in the media. In our content analysis, differences in the framing of terror-related incidents can be found between Nigeria and Burkina Faso. In Nigeria, most of these incidents are reported through an angle of oppression, violence, and intimidation against civilians for the purpose of political gains, whereas in Burkina Faso most terror-related incidents are reported as the brutal slaughter of many oppressed people. This can be observed in Figure 7. Furthermore, refugee-related topics, such as disasters, famine, or harassment against the LGBTQ (Lesbian, Gay, Bisexual, Transgender, or Queer people) community in camps are reported only in very few news articles. These topics are either considered not newsworthy, or these events are reported with a different focus.

Such low and biased media coverage of refugee crises can have a significant impact on public perception and policy decisions. In light of the findings of this study, several policy recommendations can be derived to address this issue. Above all, governments and international organizations should encourage media outlets to provide diverse and inclusive coverage of refugee crises. This can be achieved through subsidies or tax breaks for media outlets that prioritize covering underrepresented groups, including refugees. Media outlets should also be transparent about their reporting and editorial practices, including how they select stories and sources. They should also be held accountable for any inaccurate or biased reporting. These improvements can be achieved to a great extent by educating the journalists. Journalists should receive training on how to report on refugee crises in a fair and accurate manner. This could include training on cultural sensitivity, the history of displacement, and the international legal framework governing refugees. Furthermore, to accurately cover stories, governments and international organizations should facilitate access for journalists to refugee camps and other areas where refugees are located. This can help to ensure that the voices of refugees are heard and that their experiences are accurately represented in media coverage. Governments and international organizations should furthermore support independent media outlets that are committed to providing fair and accurate coverage of refugee crises. This can help to counter the influence of media outlets that prioritize sensationalism and bias over factual reporting.

While insights and elaborations on the spatial and thematic coverage of news media presented in this paper are based on a limited data set that is curated from 2017 news articles for a period of only 6 months, it certainly draws attention to interesting patterns of news reporting in this region. Future research would build up on these findings and focus on two aspects: the spatial associations of news reporting on the refugee crisis in this region will be further explored, and the news articles will be re-explored through a linguistic perspective to systematically determine the various framings of the news corresponding to different regions in West Africa. To facilitate these research directions, more news articles will be harvested and curated, and the WARRe dataset will be expanded beyond its current time period.

## 6. Conclusions

A total of 2017 news articles related to the refugee crisis, gathered over a 6-month period, were manually coded to investigate the world news coverage of the refugee crisis in West Africa. A content analysis was conducted to identify the predominant themes appearing in the news articles. As a result, 505 unique locations were found in association with 42 distinct themes found in the news articles. A comparative analysis with the UNHCR repository of refugee-related locations revealed that 96.8% of refugee-related locations in West Africa were not covered in the news during the time frame considered in this study. This raises an important question, as to what the locations that *are* mentioned in the media represent, if not official refugee-related locations? Are these locations hosting refugees, but are undocumented in the official databases? The coded dataset revealed that the themes reported vary across the different locations in the region, indicating the many facets of the crisis. The themes *Development aid*, *Political statement*, and *Relocation* are the most widely discussed topics during the considered time period, and these topics are reported

mostly in association with locations in Nigeria. Evidence shows a focus on the ongoing conflict in north-eastern Nigeria (including regions adjacent to the Lake Chad Basin), whilst incidents in Burkina Faso are rather neglected. These findings should call for regional and international policy revisions and recommendations on journalism and media coverage.

To identify factors that influence news reporting in the West African region, a linear multiple regression analysis was conducted with several covariates. The results of this analysis conclude that *GDP per capita, Regulatory quality of governments, Political stability of countries, Annual GDP, Control of corruption, Population, the Fulani ethnic population, and the Refugee population* of a country (in order) can predict the news coverage of the countries in West Africa with significance. These findings align with those that were derived for news coverage of crisis situations in other regions of the world. Underlying mechanisms for the influence of such factors should be further studied in detail in future research.

The spatial point of view to news reporting, and various themes associated with space found in this study, add significant value to the current state of the art. The methodology introduced in this study for data curation can be followed for similar studies covering crises in other parts of the world, thereby increasing the data available to identify news reporting biases. Better policy implementations can follow only if reliable data is available to showcase the current shortcomings in news reporting. With this study, we present an approach to quantitatively assess news coverage of a crisis in space and time, to categorize the topics of news reports, and specifically, to showcase the spatial coverage biases present in our focus region, West Africa, and therefore how "unseen" a large part of the refugee-related locations are.

**Supplementary Materials:** The following supporting information can be downloaded at: https://www.mdpi.com/article/10.3390/ijgi12040175/s1.

**Author Contributions:** The individual contributions of the authors are as follows: Conceptualization, Hansi Senaratne, Martin Mühlbauer, Ralph Kiefl, Torsten Riedlinger and Hannes Taubenböck; Data curation, Hansi Senaratne, Martin Mühlbauer, Ralph Kiefl, Andrea Cárdenas and Lallu Prathapan; Formal analysis, Hansi Senaratne; Funding acquisition, Carolin Biewer and Hannes Taubenböck; Investigation, Hansi Senaratne, Martin Mühlbauer and Ralph Kiefl; Methodology, Hansi Senaratne, Martin Mühlbauer, Ralph Kiefl and Hannes Taubenböck; Resources, Torsten Riedlinger and Hannes Taubenböck; Software, Martin Mühlbauer and Ralph Kiefl; Supervision, Hannes Taubenböck; Validation, Ralph Kiefl; Visualization, Hansi Senaratne; Writing—original draft, Hansi Senaratne, Martin Mühlbauer and Ralph Kiefl; Writing—review & editing, Hansi Senaratne, Torsten Riedlinger, Carolin Biewer and Hannes Taubenböck. All authors have read and agreed to the published version of the manuscript.

**Funding:** This study has been conducted as part of the project MIGRAWARE (Grant No. 01LG2082C), funded by the German Federal Ministry of Education and Research (BMBF)- programme WASCAL WRAP 2.0., the Megacities project of the VW Momentum Initiative, and the project Open Search @ DLR phase II (internal DLR project).

**Data Availability Statement:** The curated dataset will be available under an open-source license as soon as the paper is published.

**Conflicts of Interest:** The authors declare no conflict of interest.

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
