# Peer review of "The Unseen—An Investigative Analysis of Thematic and Spatial Coverage of News on the Ongoing Refugee Crisis in West Africa"

_ijgi, doi:10.3390/ijgi12040175_

Round 1

Reviewer 1 Report

The paper brings an investigative analysis including a survey from news about the West Africa refugee crisis. There is an important contribution about the issue and the found reveal interesting details.

However, in my opinion I’m not sure if the paper falls under IJGI scope.

The title doesn't make it clear than the paper will describe about an investigative approach about spatial coverage taking into account the crises existing. My first question: why analyze only 12 March to 15 September? Please avoid acronym at abstract – UNHCR Isn’t nor clear how the linear multiple regression was used as described in the Introduction section. In my opinion the focus of the research is conducting a curate data set analysis based “online news articles” Line 297 –dependant à dependen

tigure 8 –please change for a table

Reviewer 2 Report

Thanks for the opportunity for reviewing this paper. This paper utilized open dataset to investigate the coverage of news on the refugee crisis in West Africa. I think the topic of the article is very interesting and of practical meaningfulness. Besides, I also think the paper is very well written. That being said, I have the following suggestions for authors’ consideration:

1.      In the abstract, the authors may consider adding new sentences mentioning the existing gap. Have similar works been done by others, if so, how your work differentiate from and contribute to theirs? This can give a clearer picture about your contribution to the field.

2.      Similarly, the authors can consider discussing the gaps in the introduction section.

3.      The current introduction seems too short and makes the length of each section seemingly imbalanced. The author can consider combining the “Introduction” and “Background & related work” together into a new introduction section.

4.      The author can consider adding a table to illustrate and compare the used dataset.

5.      The authors mentioned that 2017 news are covered by the dataset. Please consider if this amount of data is enough and well-representative. If not, the author may discuss this and admit this as a limitation.

6.      The authors may discuss why specific factors, such as  GDP per Capita and Political stability are the most influential factors and point out revealing such underlying mechanism as one possible direction for future work.

7.      Well the references cover specific aspects, there are still some relevant I suggest the authors citing: “A review of human mobility research based on big data and its implication for smart city development”, “The'Good'and'Bad'Refugees? Imagined Refugeehood (s) in the Media Coverage of the Migration Crisis”.

Reviewer 3 Report

Thank you for the possibility to review the paper entitled ‘The Unseen - An Analysis of News Content and Spatial Coverage of the Refugee Crisis in West Africa’.

I think the paper is rich in information and brings some new elements on the international literature of spatial aspects of the refugee crisis in West Africa. However, the paper needs some revisions before being potentially considered for publication.

First, the introduction should better present what this paper brings new in the international literature of spatial aspects of the refugee crisis and in African refugee studies in particular.

Second, the literature review is quite short, an aspect reflected also in the paper’s reference list which has to be expanded. For instance, on the ethics of forced displacement there is the book of S. Parekh, 2016 (https://library.oapen.org/handle/20.500.12657/25886). There are also many case studies on refuges displaced to other regions – see for example the case of refugees displaced from Syria to Egypt in a study of Mansour S. published  in journal Information and Learning Science, 2018. Moreover, authors talk about relocation issues and this is highly debated in existing literature, so some more examples of relocation studies can be mentioned  (see for instance the study of Risteiu Toader N. et al, 2022 in journal Eurasian Geography and Economics). Then, there is the case of internal/domestic migrants (see Cantor et al - doi: 10.1093/rsq/hdaa016) and how internal migrants are stigmatized (see O’Brien T. et al, 2022 in journal Identities, see also the case of marginal people in multiethnic neighbourhoods – doi: 10.1080/1070289X.2021.1920774). Moreover, displaced or refugee people usually feel trauma during displacement and this can be counselled (see the study of Marotta S. in journal Counseling & Development, 2011) and trauma could be long-term transmitted to the next generations (see Varan C. et al, 2018 in journal Area). Therefore, empathy is an important sentiment for victims of past harsh politics (see doi: 10.1080/15387216.2019.1581632 and see the study of L.K. Taylor and C. Glen, 2019, in Journal of Community & Applied Social Psychology).

So more examples as the above ones could complement the existing literature review section of the paper.

Third, results and discussions are nicely presented, but some policy recommendations would be good to be addressed.

Finally, conclusions are too short. This section should include one paragraph on the international and regional/African implications of this study or how the novelty of  this study brings additional academic value to what other studies have presented by now on refugee studies.

Round 2

Reviewer 1 Report

Dear,

The authors does not show the concern to bring modification to make better for paper.

Reviewer 3 Report

The article is now much improved, so I am happy to accept this article for publication.